# Probabilistic Matrix Factorization for Automated Machine Learning

**Nicolo Fusi, Rishit Sheth**
Microsoft Research, New England
{nfusi,rishet}@microsoft.com

**Melih Elibol**[*]
EECS, University of California, Berkeley
elibol@cs.berkeley.edu

## Abstract

In order to achieve state-of-the-art performance, modern machine learning techniques require careful data pre-processing and hyperparameter tuning. Moreover, given the ever increasing number of machine learning models being developed, model selection is becoming increasingly important. Automating the selection and tuning of machine learning pipelines, which can include different data pre-processing methods and machine learning models, has long been one of the goals of the machine learning community. In this paper, we propose to solve this meta-learning task by combining ideas from collaborative filtering and Bayesian optimization. Specifically, we use a probabilistic matrix factorization model to transfer knowledge across experiments performed in hundreds of different datasets and use an acquisition function to guide the exploration of the space of possible pipelines. In our experiments, we show that our approach quickly identifies high-performing pipelines across a wide range of datasets, significantly outperforming the current state-of-the-art.

## 1 Introduction

Machine learning models often depend on hyperparameters that require extensive fine-tuning in order to achieve optimal performance. For example, state-of-the-art deep neural networks have highly tuned architectures and require careful initialization of the weights and learning algorithm (for example, by setting the initial learning rate and various decay parameters). These hyperparameters can be learned by cross-validation (or holdout set performance) over a grid of values, or by randomly sampling the hyperparameter space [2]; but, these approaches do not take advantage of any continuity in the parameter space. More recently, *Bayesian optimization* has emerged as a promising alternative to these approaches [25, 8, 16, 1, 23, 3]. In Bayesian optimization, the loss (*e.g.* root mean square error) is modeled as a function of the hyperparameters. A regression model (usually a Gaussian process) and an acquisition function are then used to iteratively decide which hyperparameter setting should be evaluated next. More formally, the goal of Bayesian optimization is to find the vector of hyperparameters $\boldsymbol{\theta}$ that corresponds to

$$\arg\min_{\boldsymbol{\theta}} \mathscr{L}(\mathcal{M}(\mathbf{x}; \boldsymbol{\theta}), \mathbf{y}),$$

where $\mathcal{M}(\mathbf{x}; \boldsymbol{\theta})$ are the predictions generated by a machine learning model $\mathcal{M}$ (*e.g.* SVM, random forest, etc.) with hyperparameters $\boldsymbol{\theta}$ on some inputs $\mathbf{x}$, $\mathbf{y}$ are the targets/labels, and $\mathscr{L}$ is a loss function. Usually, the hyperparameters are a subset of $\mathbb{R}^D$, although in practice many hyperparameters can be discrete (*e.g.* the number of layers in a neural network) or categorical (*e.g.* the loss function to use in a gradient boosted regression tree).

---

[*]Work conducted at Microsoft Research, New England

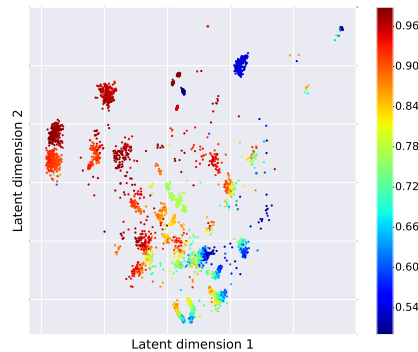

Figure 1: Two-dimensional embedding of 5,000 ML pipelines across 576 OpenML datasets. Each point corresponds to a pipeline and is colored by the AUROC obtained by that pipeline in one of the OpenML datasets (OpenML dataset id 943).

Bayesian optimization techniques have been shown to be very effective in practice and sometimes identify better hyperparameters than human experts, leading to state-of-the-art performance in computer vision tasks [23]. One drawback of these techniques is that they are known to suffer in high-dimensional hyperparameter spaces, where they often perform comparably to random search [12]. This limitation has both been shown in practice [12], as well as studied theoretically [25, 6] and is due to the necessity of sampling enough hyperparameter configurations to get a good estimate of the predictive posterior over a high-dimensional space. In practice, this is not an insurmountable obstacle when considering the fine-tuning of a handful of parameters in a single model, but it is becoming increasingly impractical as the focus of the community shifts from tuning individual hyperparameters to identifying entire ML pipelines consisting of data pre-processing methods, machine learning models, *and* their hyperparameters [4].

Our goal in this paper is indeed not only to tune the hyperparameters of a given model, but also to identify which model to use and how to pre-process the data. We do so by leveraging experiments already performed across different datasets $\mathcal{D} = \{\mathcal{D}_1, \ldots, \mathcal{D}_D\}$ to solve the optimization problem

$$\arg\min_{\mathcal{M}, \mathcal{P}, \boldsymbol{\theta}_m, \boldsymbol{\theta}_p} \mathscr{L}(\, \mathcal{M}(\, \mathcal{P}(\mathbf{x}; \boldsymbol{\theta}_p); \boldsymbol{\theta}_m), \, \mathbf{y}\,),$$

where $\mathcal{M}$ is the ML model with hyperparameters $\boldsymbol{\theta}_m$ and $\mathcal{P}$ is the pre-processing method with hyperparameters $\boldsymbol{\theta}_p$. In the rest of the paper, we refer to the combination of pre-processing method, machine learning model and their hyperparameters as an *ML pipeline*. ML pipeline space can have a combination of continuous, discrete, and categorical dimensions (*e.g.* the "model" dimension can be a choice between a random forest or an SVM), as well as encode a complex heirarchical structure with some dimensions conditioned on others (*e.g.* "the number of trees" dimension in a random forest). This mixture of types makes modeling continuity in this space particularly challenging. For this reason, unlike previous work, we consider "instantiations" of pipelines, meaning that we fix the set of pipelines ahead of training. For example, an instantiated pipeline can consist in computing the top 5 principal components of the input data and then applying a random forest with 1000 trees. Importantly, extensive experiments in Section 4 demonstrate that (i) low effective dimensionality of hyperparameter spaces suggested by [2] extends to pipeline space, and (ii) Bayesian optimization performed in these discretized spaces leads to significantly better performance than approaches that utilize continuity.

We show that the problem of predicting the performance of ML pipelines on a new dataset can be cast as a collaborative filtering problem that can be solved with probabilistic matrix factorization techniques. The approach we follow in the rest of this paper, based on Gaussian process latent variable models [10, 9], embeds different pipelines in a latent space based on their performance across different datasets. For example, Figure 1 shows the first two dimensions of the latent space of ML pipelines identified by our model on OpenML [28] datasets. Each dot corresponds to an ML pipeline and is colored depending on the AUROC achieved on a holdout set for a given OpenML

dataset. Since our probabilistic approach produces a full predictive distribution over the performance of the ML pipelines considered, we can use it in conjunction with acquisition functions commonly used in Bayesian optimization to guide the exploration of the ML pipeline space.

## 2    Related work

The concept of leveraging experiments performed in previous problem instances has been explored in different ways by two different communities. In the Bayesian optimization community, most of the work revolves around either casting this problem as an instance of multi-task learning or by selecting the first parameter settings to evaluate on a new dataset by looking at what worked in related datasets (we will refer to this as meta-learning for cold-start). In the multi-task setting, Swersky et al. (2013) [27] have proposed a multi-task Bayesian optimization approach leveraging multiple related datasets in order to find the best hyperparameter setting for a new task. For instance, they suggested using a smaller dataset to tune the hyperparameters of a bigger dataset that is more expensive to evaluate. Schilling et al. (2015) [21] also treat this problem as an instance of multi-task learning, but instead of treating each dataset as a separate task (or output), they effectively consider the tasks as conditionally independent given an indicator variable specifying which dataset was used to run a given experiment. Springenburg et al. (2016) [24] do something similar with Bayesian neural networks, but instead of passing an indicator variable, their approach learns a dataset-specific embedding vector. Perrone et al. (2017) [18] also effectively learn a task-specific embedding, but instead of using Bayesian neural networks end-to-end like in [24], they use feed-forward neural networks to learn the basis functions of a Bayesian linear regression model.

Other approaches address the cold-start problem by evaluating parameter settings that worked well in previous datasets. The most successful attempt to do so for automated machine learning problems (*i.e.* in very high-dimensional and structured parameter spaces) is the work by [4]. In their paper, the authors compute meta-features of both the dataset under examination as well as a variety of OpenML [28] datasets. These meta-features include for example the number of classes or the number of samples in each dataset. They measure similarity between datasets by computing the L1 norm of the meta-features and use the optimization runs from the nearest datasets to "warm-start" the optimization. Reif et al. (2012) [19] also use meta-features of the dataset to warm-start the optimization performed by a genetic algorithm. Feurer et al. (2015) [5] consider learning the dataset similarity function from training data for warm-starting, and Wistuba et al. (2015) [30] extend this by additionally taking into account the performance of hyperparameter configurations evaluated on the new dataset. In the same paper, they also propose to carefully pick these evaluations such that the similarity between datasets is more accurately represented, although they found that this doesn't result in improved performance in their experiments.

Other related work has been produced in the context of algorithm selection for satisfiability problems. In particular, Stern et al. (2010) [26] tackled constraint solving problems and combinatorial auction winner determination problems using a latent variable model to select which algorithm to use. Their model performs a joint linear embedding of problem instances and experts (*e.g.* different SAT solvers) based on their meta-features and a sparse matrix containing the results of previous algorithm runs. Malitsky and O'Sullivan (2014) [13] also proposed to learn a latent variable model by decomposing the matrix containing the performance of each solver on each problem. They then develop a model to project commonly used hand-crafted meta-features used to select algorithms onto the latent space identified by their model. They use this last model to do one-shot (*i.e.* non-iterative) algorithm selection. This is similar to what was done by [14], but they do not use the second regression model and instead perform one-shot algorithm selection directly.

Our work is most related to [4] in terms of scope (*i.e.* joint automated pre-processing, model selection and hyperparameter tuning), but we discretize the space and set up a multi-task model, while they capture continuity in parameter space in a single-task model with a smart initialization. Our approach is also loosely related to the work of [26], but we perform sequential model based optimization with a non-linear mapping between latent and observed space in an unsupervised model, while they use a supervised linear model trained on ranks for one-shot algorithm selection. The application domain of their model also required a different utility function and a time-based feedback model. The work of [11] constructs an acquisition function that also exploits performance on previous datasets, but their method does not do significantly better than random selection, whereas the experiments of Section 4 demonstrate that our proposed method provides significant improvement over random selection.

# 3 AutoML as probabilistic matrix factorization

In this paper, we develop a method that can draw information from *all* of the datasets for which experiments are available, whether they are immediately related (*e.g.* a smaller version of the current dataset) or not. The idea behind our approach is that if two datasets have similar (*i.e.* correlated) results for a few pipelines, it's likely that the remaining pipelines will produce results that are similar as well. This is somewhat reminiscent of a collaborative filtering problem for movie recommendation, where if two users liked the same movies in the past, it's more likely that they will like similar ones in the future.

More formally, given $N$ machine learning pipelines and $D$ datasets, we train each pipeline on part of each dataset and we evaluate it on a holdout set. This gives us a matrix $\mathbf{Y} \in \mathbb{R}^{N \times D}$ summarizing the performance of each pipeline in each dataset. For example, $\mathbf{Y}$ may represent balanced accuracies in a classification setting or RMSE in a regression setting. Having observed these performances, the task of predicting the performance of any of them on a new dataset can be cast as a matrix factorization problem.

Specifically, we are seeking a low rank decomposition such that $\mathbf{Y} \approx \mathbf{XW}$, where $\mathbf{X} \in \mathbb{R}^{N \times Q}$ and $\mathbf{W} \in \mathbb{R}^{Q \times D}$, where $Q$ is the dimensionality of the latent space. As done in [10] and [20], we consider the probabilistic version of this task, known as *probabilistic matrix factorization*:

$$p(\mathbf{Y} \,|\, \mathbf{X}, \mathbf{W}, \sigma^2) = \prod_{n=1}^{N} \mathcal{N}(\mathbf{y}_n \,|\, \mathbf{x}_n \mathbf{W}, \sigma^2 \mathbb{I}), \tag{1}$$

where $\mathbf{x}_n$ is a row of the latent variables $\mathbf{X}$ and $\mathbf{y}_n$ is a row of measured performances for pipeline $n$. In this setting both $\mathbf{X}$ and $\mathbf{W}$ are unknown and must be inferred.

## 3.1 Non-linear matrix factorization with Gaussian process priors

The probabilistic matrix factorization approach just introduced assumes that the entries of $\mathbf{Y}$ are linearly related to the latent variables. In nonlinear probabilistic matrix factorization [10], the elements of $\mathbf{Y}$ are given by a *nonlinear function* of the latent variables, $y_{n,d} = f_d(\mathbf{x}_n) + \epsilon$, where $\epsilon$ is independent Gaussian noise. This gives a likelihood of the form

$$p\left(\mathbf{Y} \,|\, \mathbf{X}, \mathbf{f}, \sigma^2\right) = \prod_{n=1}^{N} \prod_{d=1}^{D} \mathcal{N}\left(y_{n,d} | f_d\left(\mathbf{x}_n\right), \sigma^2\right), \tag{2}$$

Following [10], we place a Gaussian process prior over $f_d(\mathbf{x}_n)$ so that any vector $\mathbf{f}$ is governed by a joint Gaussian density, $p\left(\mathbf{f} \,|\, \mathbf{X}\right) = \mathcal{N}\left(\mathbf{f} | \mathbf{0}, \mathbf{K}\right)$, where $\mathbf{K}$ is a covariance matrix, and the elements $\mathbf{K}_{i,j} = k(\mathbf{x}_i, \mathbf{x}_j)$ encode the degree of correlation between two samples as a function of the latent variables. If we use the covariance function $k\left(\mathbf{x}_i, \mathbf{x}_j\right) = \mathbf{x}_i^\top \mathbf{x}_j$, which is a prior corresponding to linear functions, we recover a model equivalent to (1). Alternatively, we can choose a prior over non-linear functions, such as a squared exponential covariance function with automatic relevance determination (ARD, one length-scale per dimension), $k\left(\mathbf{x}_i, \mathbf{x}_j\right) = \alpha \exp\left(-\frac{\gamma_q}{2} ||\mathbf{x}_i - \mathbf{x}_j||^2\right)$, where $\alpha$ is a variance (or amplitude) parameter and $\gamma_q$ are length-scales. The squared exponential covariance function is infinitely differentiable and hence is a prior over very smooth functions. In practice, such a strong smoothness assumption can be unrealistic and is the reason why the Matern class of kernels is sometimes preferred [29]. In the rest of this paper we use the squared exponential kernel and leave the investigation of the performance of Matern kernels to future work.

After specifying a GP prior, the marginal likelihood is obtained by integrating out the function $f$ under the prior

$$p(\mathbf{Y} \,|\, \mathbf{X}, \boldsymbol{\theta}, \sigma^2) = \int p(\mathbf{Y} \,|\, \mathbf{X}, \mathbf{f}) \, p(\mathbf{f} \,|\, \mathbf{X}) \, d\mathbf{f} = \prod_{d=1}^{D} \mathcal{N}(\mathbf{Y}_{:,d} \,|\, \mathbf{0}, \, \mathbf{K}(\mathbf{X}, \mathbf{X}) + \sigma^2 \mathbb{I}), \tag{3}$$

where $\boldsymbol{\theta} = \{\alpha, \gamma_1, \ldots, \gamma_q\}$.

In principle, we could add metadata about the pipelines and/or the datasets by adding additional kernels. As we discuss in Section 4 and show in Figure 2 (and supplementary Figure 1), we didn't find this to help in practice since the latent variable model is able to capture all the necessary information even in the fully unsupervised setting.

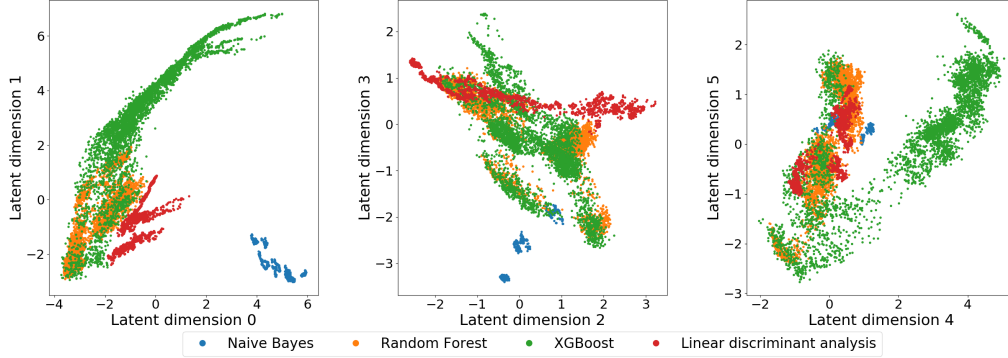

Figure 2: Latent embeddings of 42,000 machine learning pipelines colored according to which model was included in each pipeline. These are paired plots of the first 6 dimensions of our 20-dimensional latent space. The latent space effectively captures structure in the space of models.

## 3.2 Inference with missing data

Running multiple pipelines on multiple datasets is an embarrassingly parallel operation, and our proposed method readily takes advantage of these kinds of computationally cheap observations. However, in applications where it is expensive to gather such observations, $\mathbf{Y}$ will be a sparse matrix, and it becomes necessary to be able to perform inference with missing data. Given that the marginal likelihood in (3) follows a multivariate Gaussian distribution, marginalizing over missing values is straightforward and simply requires "dropping" the missing observations from the mean and covariance. More formally, we let $N_d$ denote the number of pipelines evaluated for dataset $d$, and define an indexing function $e(d) : \mathbb{N} \to \mathbb{N}^{N_d}$ that, given a dataset index $d$, returns the list of $N_d$ pipelines that have been evaluated on $d$. We can then rewrite (3) as

$$p(\mathbf{Y} \,|\, \mathbf{X}, \boldsymbol{\theta}, \sigma^2) = \prod_{d=1}^{D} \mathcal{N}(\mathbf{Y}_{e(d),d} \,|\, \mathbf{0}, \, \mathbf{C}_d), \tag{4}$$

where $\mathbf{C}_d = \mathbf{K}(\mathbf{X}_{e(d)}, \mathbf{X}_{e(d)}) + \sigma^2 \mathbb{I}$. The negative log-likelihood (NLL) of the data under this model is given by $\sum_{d=1}^{D} \mathrm{NLL}_d$, where

$$\mathrm{NLL}_d = \frac{1}{2} \Big( N_d \log(2\pi) + \log|\mathbf{C}_d| + \mathbf{Y}_{e(d),d}^{\top} \, \mathbf{C}_d^{-1} \, \mathbf{Y}_{e(d),d} \Big). \tag{5}$$

Similar to [10], we infer the parameters $(\boldsymbol{\theta}, \sigma)$ and latent variables $\mathbf{X}$ by minimizing NLL using stochastic gradient-based optimization. Specifically, we update the parameters and latent variables with a randomly selected batch of datasets $\mathcal{B}$ as

$$\boldsymbol{\theta}^{t+1} = \boldsymbol{\theta}^t - \eta \frac{D}{|\mathcal{B}|} \sum_{d \in \mathcal{B}} \frac{\partial \mathrm{NLL}_d}{\partial \boldsymbol{\theta}}, \qquad \mathbf{x}_n^{t+1} = \mathbf{x}_n^t - \eta \frac{N_n}{N_{\mathcal{B}n}} \sum_{d \in \mathcal{B}} \frac{\partial \mathrm{NLL}_d}{\partial \mathbf{x}_n}, \tag{6}$$

where $t$ denotes training iteration, $\eta$ is the learning rate, $N_n$ denotes the total number of observations for pipeline $n$, and $N_{\mathcal{B}n}$ denotes the number of observations for pipeline $n$ in the batch (we use the convention $\frac{\partial \mathrm{NLL}_d}{\partial \mathbf{x}_n} = 0$ when the entry $y_{n,d}$ is unobserved). Since our approach allows incremental re-training, existing pipeline embeddings can be updated with observations from new datasets, and new pipelines can be included given their performance observations.

## 3.3 Predictions

Predictions from the model can be easily computed by following the standard derivations for Gaussian process regression [29]. The predicted performance $y_{n,*}$ of pipeline $n$ for a new dataset $*$ is given by

$$p(y_{n,*} \,|\, \mathbf{X}, \boldsymbol{\theta}, \sigma) = \mathcal{N}(y_{n,*} \,|\, \mu_{n,*}, v_{n,*}) \tag{7}$$

$$\mu_{n,*} = \mathbf{k}_{e(*),n}^{\top} \, \mathbf{C}_*^{-1} \, \mathbf{y}_*$$

$$v_{n,*} = k_{n,n} + \sigma^2 - \mathbf{k}_{e(*),n}^{\top} \mathbf{C}_*^{-1} \mathbf{k}_{e(*),n},$$

remembering that $\mathbf{C}_* = \mathbf{K}(\mathbf{X}_{e(*)}, \mathbf{X}_{e(*)}) + \sigma^2 \mathbb{I}$ and defining $\mathbf{k}_{e(*),n} = \mathbf{K}(\mathbf{X}_{e(*)}, \boldsymbol{x}_n)$ and $k_{n,n} = \mathbf{K}(\boldsymbol{x}_n, \boldsymbol{x}_n)$.

The computational complexity for generating these predictions is largely determined by the number of pipelines already evaluated for a test dataset and is due to the inversion of a $N_* \times N_*$ matrix. This is not particularly onerous because the typical number of evaluations is likely to be in the hundreds, given the cost of training each pipeline and the risk of overfitting to the validation set if too many pipelines are evaluated.

### 3.4 Acquisition functions

The model described so far can be used to predict the expected performance of each ML pipeline as a function of the pipelines already evaluated, but does not yet give any guidance as to which pipeline should be tried next. A simple approach to pick the next pipeline to evaluate is to select the pipeline resulting in the maximum predicted performance or $\mathrm{argmax}_n \{\mu_{n,*}\}$. However, such a utility function, also known as an acquisition function, would discard information about the uncertainty of the predictions. One of the most widely used acquisition functions is expected improvement (EI) [15], which is given by the expectation of the improvement function

$$I(y_{n,*}, y_{best}) \triangleq (y_{n,*} - y_{best})\mathbb{I}(y_{n,*} > y_{best}), \qquad \mathrm{EI}_{n,*} \triangleq \mathbb{E}[I(y_{n,*}, y_{best})],$$

where $y_{best}$ is the best result observed. Since $y_{n,*}$ is Gaussian distributed (see (7)), this expectation can be computed analytically:

$$\mathrm{EI}_{n,*} = \sqrt{v_{n,*}} \left[ \gamma_{n,*} \Phi(\gamma_{n,*}) + \mathcal{N}(\gamma_{n,*} \,|\, 0, 1) \right],$$

where $\Phi$ is the cumulative distribution function of the standard normal, and $\gamma_{n,*} = \frac{\mu_{n,*} - y_{best} - \xi}{\sqrt{v_{n,*}}}$, where $\xi$ is a free parameter to encourage exploration. After computing the expected improvement for each pipeline, the next pipeline to evaluate is simply given by $\mathrm{argmax}_n (\mathrm{EI}_{n,*})$. The expected improvement is just one of many possible acquisition functions, and different problems may require different acquisition functions. See [22] for a review.

## 4 Experiments

In this section, we compare our method in a classification setting to a series of baselines that includes auto-sklearn [4], the current state-of-the-art approach and overall winner of the ChaLearn AutoML competition [7]. We ran all of the experiments on 553 OpenML [28] datasets selected by filtering for binary and multi-class classification problems with no more than $10,000$ samples and no missing values, although our method is capable of handling datasets which cause ML pipeline runs to be unsuccessful (described below).

### 4.1 Generation of training data

We generated training data for our method by splitting each OpenML dataset in 80% training data, 10% validation data and 10% test data, running $42,000$ ML pipelines on each dataset and measuring the normalized accuracy, *i.e.* accuracy rescaled such that random performance is 0 and perfect performance is 1.0.

We generated the pipelines by sampling a combination of pre-processors $\mathcal{P} = \{P^1, P^2, ..., P^n\}$, machine learning models $\mathcal{M} = \{M^1, M^2, ..., M^m\}$, and their corresponding hyperparameters $\Theta_P = \{\theta_P^1, ..., \theta_P^n\}$ and $\Theta_M = \{\theta_M^1, ..., \theta_M^m\}$ from the entries in supplementary Table 1. All the models and pre-processing methods we considered were implemented in scikit-learn [17]. We sampled the parameter space by using functions provided in the auto-sklearn library [4]. Similar to what was done in [4], we limited the maximum training time of each individual model within a pipeline to 30 seconds and its memory consumption to 16GB. Because of network failures and the cluster occasionally running out of memory, the resulting matrix $\mathbf{Y}$ was not fully sampled and had approximately 21% missing entries. As pointed out in the previous section, this is expected in realistic applications and is not a problem for our method, since it can easily handle sparse data.

Out of the 553 total datasets, 100 were identified to comprise the held-out test set. Full details including the IDs of both training and test sets are provided in the supplementary material.

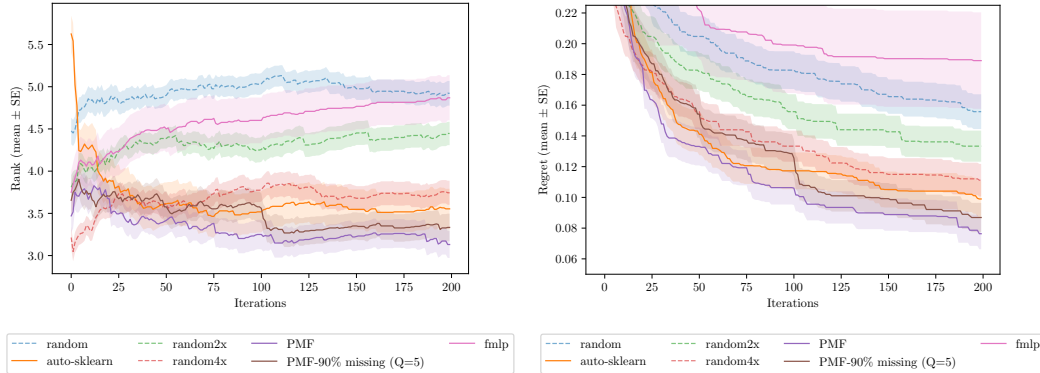

Figure 3: (Left) Average rank of all the approaches we considered as a function of the number of iterations. For each holdout dataset, the methods are ranked based on the normalized accuracy obtained on the validation set at each iteration. The ranks are then averaged across datasets. Lower is better. The shaded areas represent the standard error for each method. (Right) Difference between the maximum normalized accuracy observed on the test set and the normalized accuracy. Lower is better.

## 4.2 Parameter settings

We set the number of latent dimensions to $Q = 20$, stochastic gradient descent learning rate to $\eta = 1e^{-7}$, and (column) batch-size to 50. The latent space was initialized using PCA, and training was run for 300 epochs (corresponding to approximately 3 hours on a 16-core Azure machine). Finally, we configured the acquisition function with $\xi = 0.01^2$.

## 4.3 Results

We compared the model described in this paper, **PMF**, to the following methods:

- **random**. For each test dataset, we performed a random search by sampling each pipeline to be evaluated from the set of 42,000 at random without replacement.

- **random 2x**. Same as random, but with twice the budget. This simulates parallel evaluation of pipelines and is a strong baseline [12].

- **random 4x**. Same as random but with 4 times the budget.

- **auto-sklearn** [4]. We ran auto-sklearn for 4 hours per dataset and set to optimize normalized accuracy on a holdout set. We disabled the automated ensembling of models in order to obtain a fair comparison to the other non-ensembling methods.

- **fmlp** [21]. Factorized multi-layer perceptron as an additional baseline (implemented by us).

Our method uses the same procedure used in [4] to warm-start the process by selecting the first 5 pipelines, after which the acquisition function selects subsequent pipelines.

The left plot of Figure 3 shows the average rank for each method as a function of the number of iterations (*i.e.* the number of pipelines evaluated). Starting from the first iteration, our approach consistently achieves the best average rank. Auto-sklearn is the second best model, outperforming random 2x and almost matched by random 4x. The performance of the factorized MLP is between random and random-2x, even after tuning its hyperparameters to improve performance. This is significantly worse than both auto-sklearn and the proposed method. Please note that random 2x and random 4x are only intended as baselines that are easy to understand and interpret, but that in no way can be considered practical solutions, since they both have a much larger computational budget than the non-random methods. Additionally, we measured the difference between the maximum normalized accuracy obtained by any pipeline in each dataset and the one obtained by the pipeline

selected at each iteration. The results summarized in the right plot of Figure 3 show that our method still outperforms all the others.

We also investigated how well our method performs when fewer observations/training datasets are available. In the first variant, we ran our method in the setting where 90% of the entries in $\mathbf{Y}$ are missing[3]. The test set remains unchanged. The additional curves labeled "PMF-90% missing $(Q = 5)$" in the plots of Figure 3 demonstrate our method degrades in performance only slightly when 10% of the observations are available versus when training on 80% available observations, but still outperforms competitors, thus demonstrating very good robustness to missing data. In the second experiment, we matched the number (and the identity, for the most part) of datasets that auto-sklearn uses to initialize its Bayesian optimization procedure. The results, shown in supplementary Figure 3, confirm that our model still outperforms competing approaches.

**Including pipeline metadata.** Our approach can easily incorporate information about the composition and the hyperparameters of the pipelines considered. This metadata could, for example, include information about which model is used within each pipeline or which pre-processor is applied to the data before passing it to the model. Empirically, we found that including this information in our model didn't improve performance (shown in supplementary Figure 4). Indeed, our model is able to effectively capture most of this information in a completely unsupervised fashion, just by observing the sparse pipelines-dataset matrix $\mathbf{Y}$. This is visible in Figure 2, where we show the latent embedding colored according to which model was included in which pipeline.

## 5   Discussion

We have presented a new approach to automatically build predictive ML pipelines for a given dataset, automating the selection of data pre-processing method and machine learning model as well as the tuning of their hyperparameters. Our approach combines techniques from collaborative filtering and ideas from Bayesian optimization to intelligently explore the space of ML pipelines, exploiting experiments performed in previous datasets. We have benchmarked our approach against the state-of-the-art with a large number of OpenML datasets with different sample sizes, number of features and number of classes. Overall, our results show that our approach outperforms both the state-of-the-art as well as a set of strong baselines.

One potential concern with our method is that it requires sampling (*i.e.* instantiating pipelines) from a potentially high-dimensional space and thus could require exponentially many samples in order to explore all areas of this space. We have found this not to be a problem for three reasons. First, many of the dimensions in the space of pipelines are conditioned on the choice of other dimensions. For example, the number of trees or depth of a random forest are parameters that are only relevant if a random forest is chosen in the "model" dimension. This reduces the effective search space significantly. Second, in our model we treat every pipeline as an additional sample, so increasing the sampling density also results in an increase in sample size (and similarly, adding a dataset also increases the effective sample size). Finally, very dense sampling of the pipeline space is only needed if the performance is very sensitive to small parameter changes, something that we haven't observed in practice. If this is a concern, we advise using our approach in conjunction with traditional Bayesian optimization methods (such as [23]) to further fine-tune the parameters.

We are currently investigating several extensions of this work. First, we would like to include dataset-specific information in our model. As discussed in Section 3, the only data taken into account by our model is the performance of each method in each dataset. Similarity between different pipelines is induced by having correlated performance across multiple datasets, and ignores potentially relevant metadata about datasets, such as the sample size or number of classes. We are currently working on including such information by extending our model using additional kernels and dual embeddings (*i.e.* embedding both pipelines and dataset in separate latent spaces). Second, we are interested in using acquisition functions that include a factor representing the computational cost of running a given pipeline [23] to handle instances when datasets have a large number of samples. The machine learning models we used for our experiments were constrained not to exceed a certain runtime, but

this could be impractical in real applications. Finally, we are planning to experiment with different probabilistic matrix factorization models based on variational autoencoders.

# 6 Data and software

Data and software available at `https://github.com/rsheth80/pmf-automl/`.

## Footnotes

[2]Although not shown in this section, we also ran the proposed method with $\xi = 0$ in the acquisition function, and this did not produce any distinguishable effect in our results.

[3]From the original observation matrix, which has $\approx 20\%$ missing entries, an additional 70% are dropped uniformly at random. The model was trained with Adam using a learning rate of $10^{-2}$ and $Q = 5$, and 3 pipelines were used for warm-starting.

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
