[Supplementary Material]

# Supplementary Material for "Probabilistic Matrix Factorization for Automated Machine Learning"

| ML / PP Algorithm | Parameter | Range |
|---|---|---|
| Polynomial Features | `degree` | `[2, 3]` |
| Polynomial Features | `interaction_only` | `{False, True}` |
| Polynomial Features | `include_bias` | `{True, False}` |
| Principal Component Analysis | `keep_variance` | `[0.5, 0.9999]` |
| Principal Component Analysis | `whiten` | `{False, True}` |
| Linear Discriminant Analysis | `shrinkage` | `{None, auto, manual}` |
| Linear Discriminant Analysis | `n_components` | `[1, 250]` |
| Linear Discriminant Analysis | `tol` | `[1e-05, 0.1]` |
| Linear Discriminant Analysis | `shrinkage_factor` | `[0.0, 1.0]` |
| Extreme Gradient Boosting | `max_depth` | `[1, 10]` |
| Extreme Gradient Boosting | `learning_rate` | `[0.01, 1.0]` |
| Extreme Gradient Boosting | `n_estimators` | `[50, 500]` |
| Extreme Gradient Boosting | `subsample` | `[0.01, 1.0]` |
| Extreme Gradient Boosting | `min_child_weight` | `[1, 20]` |
| Quadratic Discriminant Analysis | `reg_param` | `[0.0, 10.0]` |
| Extra Trees | `criterion` | `{gini, entropy}` |
| Extra Trees | `max_features` | `[0.5, 5.0]` |
| Extra Trees | `min_samples_split` | `[2, 20]` |
| Extra Trees | `min_samples_leaf` | `[1, 20]` |
| Extra Trees | `bootstrap` | `{True, False}` |
| Decision Tree | `criterion` | `{gini, entropy}` |
| Decision Tree | `max_depth` | `[0.0, 2.0]` |
| Decision Tree | `min_samples_split` | `[2, 20]` |
| Decision Tree | `min_samples_leaf` | `[1, 20]` |
| Gradient Boosted Decision Trees | `learning_rate` | `[0.01, 1.0]` |
| Gradient Boosted Decision Trees | `n_estimators` | `[50, 500]` |
| Gradient Boosted Decision Trees | `max_depth` | `[1, 10]` |
| Gradient Boosted Decision Trees | `min_samples_split` | `[2, 20]` |
| Gradient Boosted Decision Trees | `min_samples_leaf` | `[1, 20]` |
| Gradient Boosted Decision Trees | `subsample` | `[0.01, 1.0]` |
| Gradient Boosted Decision Trees | `max_features` | `[0.5, 5.0]` |
| K Neighbors | `n_neighbors` | `[1, 100]` |
| K Neighbors | `weights` | `{uniform, distance}` |
| K Neighbors | `p` | `{1, 2}` |
| Multinomial Naive Bayes | `alpha` | `[0.01, 100.0]` |
| Multinomial Naive Bayes | `fit_prior` | `{True, False}` |
| Support Vector Machine | `C` | `[0.03125, 32768.0]` |
| Support Vector Machine | `kernel` | `{rbf, poly, sigmoid}` |
| Support Vector Machine | `gamma` | `[3.05176e-05, 8.0]` |
| Support Vector Machine | `shrinking` | `{True, False}` |
| Support Vector Machine | `tol` | `[1e-05, 0.1]` |
| Support Vector Machine | `coef0` | `[-1.0, 1.0]` |
| Support Vector Machine | `degree` | `[1, 5]` |
| Random Forest | `criterion` | `{gini, entropy}` |
| Random Forest | `max_features` | `[0.5, 5.0]` |
| Random Forest | `min_samples_split` | `[2, 20]` |
| Random Forest | `min_samples_leaf` | `[1, 20]` |
| Random Forest | `bootstrap` | `{True, False}` |
| Bernoulli Naive Bayes | `alpha` | `[0.01, 100.0]` |
| Bernoulli Naive Bayes | `fit_prior` | `{True, False}` |

Table 1: List of preprocessing methods, ML models/algorithms and parameters considered.

# 1 Generation of training data

We found that some of the OpenML datasets are so easy to model, that most of the machine learning pipelines we tried worked equally well. Since this could swamp any difference between the different methods we were evaluating, we chose our test set taking into consideration the difficulty of each dataset. We did so by randomly drawing without replacement each dataset with probabilities proportional to how poorly random selection performed on it. Specifically, for each dataset, we ran random search for 300 iterations and recorded the regret. The probability of selecting a dataset was then proportional to the regret on that dataset, averaged over 100 trials of random selection. 100 datasets were selected for the test set of which 11 had been used to train auto-sklearn. Moving these 11 from test to train resulted in a training set consisting of 464 datasets. The following is the list of OpenML dataset IDs used to train the proposed method in the main paper:

```
[3, 6, 10, 11, 12, 14, 16, 18, 20, 21, 22, 26, 28, 30, 31, 32, 36, 39, 41,
43, 44, 46, 50, 54, 59, 60, 61, 62, 151, 155, 161, 162, 164, 180, 181, 182,
183, 184, 187, 189, 209, 223, 225, 227, 230, 275, 277, 287, 292, 294, 298,
300, 307, 310, 312, 313, 329, 333, 334, 335, 336, 338, 339, 343, 346, 375,
377, 383, 385, 386, 387, 389, 391, 392, 395, 400, 401, 444, 446, 448, 450,
457, 458, 461, 462, 463, 464, 465, 467, 468, 469, 472, 476, 477, 478, 479,
480, 679, 682, 685, 694, 713, 715, 716, 717, 718, 719, 720, 721, 722, 723,
725, 727, 728, 729, 730, 732, 734, 735, 737, 741, 742, 743, 744, 745, 746,
747, 748, 749, 751, 752, 754, 755, 756, 758, 759, 761, 762, 765, 766, 767,
768, 769, 770, 772, 775, 776, 777, 778, 779, 780, 782, 784, 785, 787, 788,
790, 791, 792, 793, 794, 795, 796, 797, 801, 803, 804, 805, 807, 808, 811,
813, 814, 815, 816, 817, 818, 819, 820, 821, 823, 824, 827, 828, 829, 830,
832, 833, 834, 835, 837, 841, 843, 845, 846, 847, 848, 849, 850, 853, 855,
857, 859, 860, 863, 864, 865, 866, 867, 868, 870, 871, 872, 873, 874, 875,
877, 878, 879, 880, 881, 882, 884, 885, 886, 889, 890, 892, 894, 895, 900,
901, 903, 905, 910, 912, 913, 914, 915, 916, 917, 919, 921, 922, 923, 924,
925, 928, 932, 933, 934, 935, 936, 937, 938, 941, 942, 943, 946, 947, 950,
951, 952, 953, 954, 955, 956, 958, 959, 962, 964, 965, 969, 970, 971, 973,
974, 976, 977, 978, 979, 980, 983, 987, 988, 991, 994, 995, 997, 1004, 1005,
1006, 1009, 1011, 1013, 1014, 1015, 1016, 1019, 1020, 1021, 1022, 1025,
1026, 1038, 1040, 1041, 1043, 1044, 1045, 1046, 1048, 1055, 1056, 1059,
1060, 1061, 1062, 1063, 1064, 1065, 1066, 1068, 1069, 1075, 1079, 1081,
1082, 1104, 1106, 1107, 1115, 1116, 1120, 1121, 1122, 1123, 1124, 1125,
1126, 1127, 1129, 1131, 1132, 1133, 1135, 1136, 1137, 1140, 1141, 1143,
1144, 1145, 1147, 1148, 1149, 1150, 1151, 1152, 1153, 1154, 1155, 1156,
1157, 1158, 1160, 1162, 1163, 1165, 1167, 1169, 1217, 1236, 1237, 1238,
1413, 1441, 1442, 1443, 1444, 1446, 1448, 1449, 1450, 1451, 1452, 1454,
1455, 1457, 1459, 1460, 1464, 1467, 1471, 1475, 1481, 1482, 1486, 1488,
1489, 1496, 1498, 1500, 1501, 1505, 1507, 1508, 1509, 1510, 1516, 1517,
1519, 1520, 1527, 1528, 1529, 1530, 1531, 1532, 1533, 1534, 1535, 1536,
1537, 1538, 1539, 1540, 1541, 1542, 1544, 1545, 1546, 1556, 1557, 1561,
1562, 1563, 1564, 1565, 1567, 1568, 1569, 4134, 4135, 4153, 4340, 4534,
4538, 40474, 40475, 40476, 40477, 40478, 1050, 1067, 740, 398, 23, 1036,
1049, 799, 822, 904, 806]
```

With respect to the selected held-out test set, auto-sklearn failed to complete 200 iterations on the OpenML datasets with IDs [8, 197, 279, and 1472], and the fmlp failed on the OpenML dataset with ID 887. After filtering these datasets, the final test set contained 84 OpenML datasets with the IDs:

```
[733, 812, 731, 929, 1600, 475, 726, 197, 394, 1472, 1159, 763, 1483, 1080,
836, 851, 911, 459, 37, 927, 887, 783, 1012, 764, 714, 285, 1117, 384, 888,
1447, 1100, 789, 48, 1054, 1164, 838, 869, 931, 876, 1073, 1071, 750, 1518,
948, 736, 896, 1503, 278, 279, 908, 724, 996, 891, 926, 337, 909, 826, 800,
1487, 1512, 945, 825, 949, 753, 774, 906, 902, 1473, 8, 862, 920, 1078, 683,
```

1084, 1412, 53, 276, 1543, 907, 397, 918, 771, 773, 1077, 1453, 893, 1513, 388]

## 2   Unsupervised learning capability

On a finer scale, the latent space can also capture different settings of an individual hyperparameter. After training, we plotted the latent embeddings for pipelines using PCA as a pre-processor in Figure 1, where each pipeline is embedded in a 2-dimensional space and colored by the value of the hyperparameter of interest, in this case the percent of variance retained by a PCA preprocessor. We stress that this learning capability is completely unsupervised.

Figure 1: Latent embedding of all the pipelines in which PCA is included as a pre-processor. Each point is colored according to the percentage of variance retained by PCA (*i.e.* the hyperparameter of interest when tuning PCA in ML pipelines).

## 3   Pipeline evaluation and uncertainty

We investigated how quickly our model is able to improve its predictions as more pipelines are evaluated. Figure 2a shows the mean squared error computed across the test datasets as a function of the number of evaluations. As expected the error monotonically decreases and appears to asymptote after 200 iterations. Figure 2b shows the uncertainty of the model (specifically, the posterior variance) as a function of the number of evaluations. Overall, Figure 2 a and b support that as more evaluations are performed, the model becomes less uncertain and the accuracy of the predictions increases.

Figure 2: (a) Mean squared error (MSE) between predicted and observed balanced accuracies in the test set as a function of the number of iterations. Lower is better. MSE is averaged across all test datasets. (b) Posterior predictive variance as a function of the number of iterations and averaged across all test datasets. Shaded area shows two standard errors around the mean.

## 4 Additional sparsity experiment

As an additional experiment, we trained the proposed method on 93 of the datasets used to train auto-sklearn and an additional 47 datasets selected uniformly at random from the training set of the main paper. The model and training parameters were 20 latent dimensions and a learning rate of $10^{-5}$. The evaluation was performed on the same held out test set. Figure 3 shows the outcome of this experiment and demonstrate that our method still outperforms auto-sklearn. For this experiment, the list of OpenML dataset IDs used to train our method was

```
[3, 6, 12, 14, 16, 18, 21, 22, 26, 28, 30, 31, 32, 36, 44, 46, 60, 180, 181,
182, 184, 300, 389, 391, 392, 395, 401, 679, 715, 718, 720, 722, 723, 727,
728, 734, 735, 737, 741, 743, 751, 752, 761, 772, 797, 803, 807, 813, 816,
819, 821, 823, 833, 837, 843, 845, 846, 847, 849, 866, 871, 881, 901, 903,
910, 912, 913, 914, 917, 923, 934, 953, 958, 959, 962, 971, 976, 977, 978,
979, 980, 991, 995, 1019, 1020, 1021, 1040, 1041, 1056, 1068, 1069, 1116,
1120, 310, 1132, 685, 824, 1015, 1541, 50, 890, 1014, 1446, 747, 875, 1459,
721, 900, 878, 1236, 40478, 1562, 1079, 1496, 1449, 988, 796, 162, 811,
1145, 776, 457, 476, 1482, 1529, 1127, 952, 740, 1043, 1546, 4135, 1022,
853, 1237, 758, 827, 814, 450, 155, 462]
```

## 5 Including pipeline meta-data

Here, we include pipeline meta-data consisting of which pre-processor and model is used within each pipeline. The information is one-hot encoded and supplied via a linear mean function whose weights are learned during training. Figure 4 shows that including this information didn't improve performance in the case of the 80% observed matrix. However, we believe meta-data can be more beneficial for higher sparsity levels.

## 6 Increasing sparsity level, altering training update rule, and more iterations

Here, we include additional results from (i) further increasing the level of sparsity to a 0.5% observed performance matrix, (ii) utilizing Adam with learning rate $10^{-2}$ to train in the 80%-observed case, and (iii) letting methods run to 1000 total iterations. The sparsity level of 0.5% was achieved by dropping 79.5% of all observations uniformly at random. For training, we utilized Adam with learning rate $10^{-2}$ and a latent dimensionality of $Q = 2$. Warm-starting could not be used since

Figure 3: (Left) Average rank of all the approaches we considered as a function of the number of iterations. For each holdout dataset, the methods are ranked based on the balanced accuracy obtained on the validation set at each iteration. The ranks are then averaged across datasets. Lower is better. The shaded areas represent the standard error for each method. (Right) Difference between the maximum balanced accuracy observed on the test set and the balanced accuracy. Lower is better.

Figure 4: See Figure 3 for axes descriptions. "RBF-20-L1-5" in green denotes the results of the method of the main paper, and "oh-RBF-20-L1-5" in red denotes the results of including pipeline meta-data.

there were very few pipelines with multiple observations at this sparsity level; thus, a random initial pipeline was selected. We experimented with both an RBF and linear kernel. All other training settings remained unchanged from those described in the main paper. From Figure 5, we see that (i) training with a linear kernel under 99.5% sparsity produces performance between random-1x and random-2x, (ii) utilizing Adam instead of SGD still yields performance better than random-4x, and (iii) the performance relationships between the methods remains essentially the same when run for more iterations.

# 7 Additional results

The outcome of running auto-sklearn for 1000 iterations and any results from additional benchmarks will be posted to `https://github.com/rsheth80/pmf-automl/`.

Figure 5: See Figure 3 for axes descriptions. "PMF" denotes the method of the main paper (run on the 80%-observed matrix). "PMF-90% missing ($Q = 5$)" denotes the sparse variant run on the 10%-observed matrix (also described in main paper).