[Reviews · NeurIPS 2018]

Reviewer 1



This work presents an approach to deal with Automatic Machine Learning, optimizing not only the hyperparameters of a Machine Learning model but also different machine learning models to choose the best one, what it calls pipelines. This idea is implemented in Autoweka (SMAC), and also TPE can handle this task. In order to do so, it uses Probabilistic Matrix Factorization. Related to: Gaussian Process priors over this technique are used to predict the performance of pipelines in a new dataset. Mnih, Andriy, and Ruslan R. Salakhutdinov. "Probabilistic matrix factorization." Advances in neural information processing systems. 2008. The AutoML task is already solved: Kotthoff, Lars, et al. "Auto-WEKA 2.0: Automatic model selection and hyperparameter optimization in WEKA." The Journal of Machine Learning Research 18.1 (2017): 826-830. The surrogate model of AutoWeka is SMAC, which is proven to be a robust (and simple!) solution to this problem. Strengths: -> Clever use of Probabilistic Matrix Factorization to solve this task. -> Experiments are performed in a plethora of datasets. -> Good ideas for further work. Weaknesses: -> Contribution is low. No new models are presented. The contribution is employing Probabilistic Matrix Factorization to solve this problem. -> Clarity and organization. I think that this paper may be hard to read if you are not an expert of the area. Sometimes, variables are not provided an intuitive interpretation. -> The title is misleading. I would suggest to change it to Automatic Machine Learning via Collaborative Filtering. High-dimensional BO can exist in a problem with 10000 variables for example. -> In the experiments, some "hyper-hyperparameters" of this approach are given static values without justification. -> I think that this is a work in progress, at least not complete for a NIPS conference level. Many ideas remain to be implemented. Does this submission add value to the NIPS community? : It has the potential, but not yet. I think that, besides the contribution is not as high as providing a new model, it is clever. Some of the presented further work must be added to this paper. I would highlight the addition of specific information of the datasets to the model. Quality: Is this submission technically sound?: Not at all, borderline at least. Although it is clever to use Probabilistic Matrix Factorization, this task is already solved and only having taken into account the accuracy and not other features is a limitation. By adding these features that other approaches do not take into account, the soundness of the paper would be higher. Are claims well supported by theoretical analysis or experimental results?: No theoretical analysis is provided and experiments support well the results talking about quantity but I am not so sure about having set some of the variables of the experiments to a fixed value (Q=20, LR = 10-7, batch-size = 50). Is this a complete piece of work or work in progress?: Work in progress. Further work, really well explained, in the conclusions section must be presented in order for this paper to be accepted. Are the authors careful and honest about evaluating both the strengths and weaknesses of their work?: I miss more analysis and explanations about the strenghts and weaknesses of the approach. Some are given, but it is not enough. Clarity: Is the submission clearly written?: Not at all. This paper is hard to read for newcomers or even researchers that are not experts. Some areas are given too much detail (for example the explanation of the EI acquisition function, I would just cite it). With that gained space, I will clearly explain what does \mathbf{X}, \mathbf{x}_i and \mathbf{x}_j mean. There are also some quantities that are not explained, as const in \mathbf{NLL}, the implications of \eta, the learning rate, but above all, in the experiments section, an interpretation of Q, \eta and batch size values chosen must be provided. Is it well organized?: I would change some things to improve organization. Regarding the abstract, the first paragraph, until "learning community" is the standard literature of Bayesian Optimization that applies to all BO papers. In this case, all this information that does not describe this particular paper represents more than half of the abstract. Also, the other paragraph lacks cohesion with the first one. Regarding introduction, the third paragraph "Bayesian optimization techniques" should be a continuation of the first one, for coherence. Other critical problem is that, based on the title, I expected a complete different thing in the paper until I read from line 44 on advance. This information should be stated earlier or the title should be changed. I encourage to rewrite the abstract to make clear the pipeline topic. Another issue, section 3.1 is not expected when you finished reading section 3. I expected what is explained in section 3.2. I would change the last paragraph of section 3, introducing section 3.1, to add cohesion. Does it adequately inform the reader?: By solving the mentioned issues, I think so. Originality: Are the tasks or methods new?: No. The methodology is built from well-known techniques. Is the work a novel combination of well-known techniques?: Yes, Probabilistic Matrix Factorization, Gaussian Processes and ML algorithms. Is it clear how this work differs from previous contributions?: Yes, I have not read nothing similar nor the related work presents nothing like the contribution of the authors. Is related work adequately cited?: Yes, the authors did a good job here. Significance: Are the results important?: Currently not. But, if more features are added of dataset, it could be interesting to analyze that results. Are others likely to use the ideas or build on them?: Same answer as before. Does the submission address a difficult task in a better way than previous work?: Not in my opinion. I prefer the SMAC methodology. But I admit that this opinion can be subjective. I prefer that the model can handle all and here the addition of Probabilistic Matrix Factorization adds more complexity. Does it advance the state of the art in a demonstrable way?: Currently not. I am not sure about the fixed values of the experiments and the comparison with others. Does it provide unique data, unique conclusions about existing data, or a unique theoretical or experimental approach?: It does provide some unique conclusions, but further work must be implemented. Arguments for acceptance: -> New methodology based on GPs and Probabilistic Matrix Factorization for Automatic Machine Learning. Arguments against acceptance: -> It is just a combination of well-known techniques that does not solve a new problem and its potential is not fully developed (further work). -> I would add more experiments. -> Title is misleading. -> Clarity and Quality are poor. Issues must be corrected. Line 138. Y -> \mathbf{Y} -> Abstract, line 14: Name the state of the art techniques. -> Abstract, line 12: Define pipeline. -> Introduction, line 22: That is an assumption that you are considering. Define that fact as an assumption, parameter space should be formalized. -> Equation of line 27: Put a number and \mathbf{x} is a matrix, so capital letter. -> The instatiations of the pipelines that you are considering are a restriction for the experiments wrt the theory. State that this is a constraint. -> Line 57: Demonstrate empirically. -> Line 119: That "idea" is a hypothesis that you are considering. Define that idea as a hypothesis. -> Line 153: Why do you choose the Squared Exponential Kernel over the Matern kernel if the Matern kernel is preferred? -> Complexity analysis: How does this methodology scale wrt pipelines/datasets? To sum up, I would consider giving a borderline to this paper if all these demands are satisfied. If there are not satisfied I can not recommend this paper although I liked the idea of using Probabilistic Matrix Factorization.

Reviewer 2



The paper 'Probabilistic Matrix Factorization for Automated Machine Learning' proposes to use a Gaussian process latent variable (GPLV) model together with the expected improvement criterion to perform algorithm selection from a finite set of machine learning pipelines by transferring knowledge from previously evaluated datasets. GPLV is fitted on a large set of meta datasets via stochastic gradient descent, and the latent variables describing the different machine learning algorithms are then used inside a Gaussian process to predict the performance of a known, but not yet evaluated configuration on a new dataset (given that there are a few observations already). In a comparison against Auto-sklearn, Random Search, 2x Random Search, 4x Random Search and factorized multi-layer perceptrons the paper demonstrates that the proposed method is indeed capable of selecting algorithms which perform better than the algorithms selected by the competitors on a large set of 100 datasets. Novely: The idea of using non-linear matrix factorization for Bayesian optimization is novel, as well as its application to Automated Machine Learning. Clarity: The paper is written very clearly and easy to understand. However, the authors should not use references like [12] as nouns, but rather write 'Malitsky and O'Sullivan (2014)'. Correctness: The paper appears to be correct except for a few very minor things which should be rather easy to fix. * The paper misses a discussion of 'Selecting Classification Algorithms with Active Testing' (Leite et al., 2012), which tackles the problem of selecting an algorithm from a set of algorithms given function evaluations on the same algorithms on a large meta dataset. While the proposed paper uses pipelines as algorithms in contrast to Leite et al. (2012), I do not think that this inhibits the applicability of active testing as a competitor. * In the 2nd paragraph the authors might want to mention 'Learning Hyperparameter Optimization Initializations' by Wistuba et al., which is the most advanced method to warm-start Bayesian optimization. * Section 3.3, shouldn't it be \bm{x}_m instead of \bm{X}_m to highligh the fact that these are vectors, not matrices? * Section 4.1, the definition of balanced accuracy appears to be incorrect. The balanced accuracy is the average of the class-wise accuracy (see [6]). Methodology: The paper uses a scalable and probabilistic method for continuous algorithm selection which is interesting and novel. The limitation to a fixed set of machine learning pipelines is well motivated and should only hurt mildly in practice. * The biggest methodological drawback is the fact that the paper uses a total of 42000 * 464 * 0.79 or 42000 * 464 * 0.1 previous function evaluations to make the system work. This is quite a lot and might hamper the practical usefulness of the proposed method. It would be great if the authors would study the impact of increasing the sparsity of the observation matrix to a level which is realistic in practice (for example 200 as 200 runs are conducted in the experiments). * Currently, the paper presents the usefulness of the model on a single benchmark. It would be great if the authors could run their method on a different benchmark, too, such as the ones given by Schilling et al. (2015) for which the required data for the proposed method is available online. * This is not relevant for the rebuttal, but would be interesting if the paper is accepted: how do the different methods compare in the long run, for example after 500 or 1000 function evaluations? Further: * Is the fmlp implementation the original implementation of [20]? Overall, the paper is well-written and proposes a novel method which is of interest to the AutoML community and should be accepted. Minor: * arg min is typeset in a suboptimal way on the first page (it is better on the 2nd). * line 50, it seems there should be the word 'hyperparameter' before the bracket * line 49, I suggest to write 'the hyperparameter search dimensions' to make it clear what dimensions are discussed here. * When presetnig two methods next to each other, one legend should be sufficient (for example in Figure 3). Comments after the rebuttal: Dear authors, thank you very much for the answers. I am looking forward to seeing bullet point #2 and bullet point #3 being addressed, either in a NIPS publication or a resubmission. I do not see a reason to change my score as answers to my points are deferred to the future.

Reviewer 3



This paper presents a method to perform hyperparameter tunning and model selection for autoML based on Bayesian optimization. The novelty relies in the use of collaborative filtering within the surrogate model to transfer knowledge between different datasets and pipelines. The idea is simple, but really interesting for practical applications. I can see how this method is used in practice, with the only limitation being the vast cost of running such large number of ML pipelines with different datasets. The paper is clearly written and easy to follow, although the title can be misleading. The rest of the paper seems to focus on the multi-task component of the contribution -which I personally find more interesting- that on the high dimensionality of the input. Besides, this is not a paper about general high-dimensional BO, because the collaborative filtering part, at least in the form it is presented in the paper, is limited to autoML applications. Although the idea could be extended to other applications, the transference (ironically) would not be trivial and depending on the application. The experiments justify the proposed method. The figures are sometimes hard to read, even zooming the PDF. It seems that the authors are using bitmaps instead of vectorial images. ---- I agree with other reviewers that the paper is a combination of existing methodologies. However, it is new and sound within the context of Bayesian optimization and the application seems very strong for the autoML community. Thus, I believe this paper is interesting for the NIPS community.